# Molecular Repolarisation of Tumour-Associated Macrophages

**DOI:** 10.3390/molecules24010009

**Published:** 2018-12-20

**Authors:** Floris J. van Dalen, Marleen H. M. E. van Stevendaal, Felix L. Fennemann, Martijn Verdoes, Olga Ilina

**Affiliations:** 1Department of Tumor Immunology, Radboud Institute for Molecular Life Sciences, Radboud University Medical Centre, 6525 GA Nijmegen, The Netherlands; Floris.vanDalen@radboudumc.nl (F.J.v.D.); Felix.L.Fennemann@radboudumc.nl (F.L.F.); Olga.Ilina@radboudumc.nl (O.I.); 2Department of Bio-Organic Chemistry, Institute for Complex Molecular Systems, Eindhoven University of Technology, 5600 MB Eindhoven, The Netherlands; M.H.M.E.v.Stevendaal@tue.nl

**Keywords:** tumour-associated macrophages, repolarisation, cancer immunotherapy, tumour microenvironment, small molecules

## Abstract

The tumour microenvironment (TME) is composed of extracellular matrix and non-mutated cells supporting tumour growth and development. Tumour-associated macrophages (TAMs) are among the most abundant immune cells in the TME and are responsible for the onset of a smouldering inflammation. TAMs play a pivotal role in oncogenic processes as tumour proliferation, angiogenesis and metastasis, and they provide a barrier against the cytotoxic effector function of T lymphocytes and natural killer (NK) cells. However, TAMs are highly plastic cells that can adopt either pro- or anti-inflammatory roles in response to environmental cues. Consequently, TAMs represent an attractive target to recalibrate immune responses in the TME. Initial TAM-targeted strategies, such as macrophage depletion or disruption of TAM recruitment, have shown beneficial effects in preclinical models and clinical trials. Alternatively, reprogramming TAMs towards a proinflammatory and tumouricidal phenotype has become an attractive strategy in immunotherapy. This work summarises the molecular wheelwork of macrophage biology and presents an overview of molecular strategies to repolarise TAMs in immunotherapy.

## 1. Introduction

A tumour-promoting microenvironment plays an important role in tumour development and hampers the onset of an effective immune response against tumour cells. The TME consists of extracellular matrix, an expanding vasculature, stromal cells and various types of immune cells. TAMs cover the dominant portion of these immune cells and can account for up to 30% of the tumour mass [1]. TAMs are responsible for the onset of a tumour-promoting, smouldering inflammation, which prevents T cells and NK cells from attacking the tumour [2]. Instead, TAMs enhance tumour development into malignancy and are generally associated with a poor prognosis [3]. This review summarises the molecular mechanisms of macrophage polarisation and discusses recent developments in TAM repolarisation strategies, their associated mechanisms and therapeutic efficacy. An overview of ‘M2 to M1′ macrophage reprogramming molecules is presented in Table 1.

## 2. Macrophages in Tumourigenesis

Macrophages are immune cells derived from the myeloid lineage and play an important role in tissue development, homeostasis and immune surveillance [4]. Mirroring the paradigm of T_h_1/T_h_2 lymphocytes, macrophages can be subdivided into two polarised subsets; classically activated macrophages and alternatively activated macrophages, M1 and M2, respectively. Macrophages polarise along the type 1 pathway in response to T_h_1-derived cytokines as interferon gamma (IFNγ), colony stimulating factor 2 (CSF2 or GM-CSF) or toll-like receptor (TLR) agonism [5,6]. This induces expression of major histocompatibility complex II (MHC II) and costimulatory molecules (CD80 and CD86) leading to efficient T cell priming. M1 cells produce nitric oxide (NO) and reactive oxygen species (ROS) associated with microbicidal and tumouricidal activity. Furthermore, M1 polarised macrophages recruit new T_h_1 cells via chemokines CXCL9 and CXCL10 and produce proinflammatory cytokines as tumour necrosis factor alpha (TNFα) and interleukins (IL)-1β, IL6, IL12, and IL23 [7,8]. M1 infiltration in tumours is associated with a good prognosis in some cancers [9,10]. On the other end of the spectrum are M2 macrophages. This immunosuppressive state is initiated by T_h_2-derived cytokines such as IL4, IL10, IL13, transforming growth factor beta (TGFβ), prostaglandin E2 (PGE2) or colony stimulating factor 1 (CSF1 or M-CSF) [10,11]. M2 macrophages lose their antigen presenting capabilities and are involved in tissue remodelling, debris scavenging and immune modulation [4]. They support angiogenesis by secretion of adrenomedullin and vascular epithelial growth factors (VEGFs), and express immunosuppressive molecules such as IL10, programmed death ligand 1 (PD-L1) and transforming growth factor beta (TGFβ) [12,13].

This simplified M1/M2 definition provides a useful framework to classify macrophage function in macrophage biology. However, this definition overlooks the spectrum of trophic/developmental macrophages involved in tissue repair, angiogenesis or bone remodelling that can be found in normal physiology [12,14,15]. During tumour development at least six distinct TAM populations can be found, often sharing features of both types, M1 and M2 [2,6,16]. This spectrum of activation states is controlled by the cytokine profile of the TME, where different cytokines can integrate and induce a broad and dynamic range of functions (Figure 1). Qian and Pollard reasoned that these distinctive phenotypes represent individual stages of tumour development; starting from tumour proliferation and invasion towards more malignant processes as vascularisation and metastasis [2].

The onset of tumourigenesis is often accompanied by a non-resolving inflammation. Central in this process are tissue-resident macrophages (F4/80^hi^), such as Kupffer cells, Langerhans cells, microglia etc., that co-evolve with the tumour [1,2]. They provide a mutagenic environment by producing nitrogen or oxygen radicals and secrete a variety of growth factors [17,18,19]. Following the development of an intraepithelial lesion, this macrophage phenotype is shifted towards an immunosuppressive subset by tumour-derived cytokines as IL4, IL10 and IL13. Bone marrow-derived monocytes (F4/80^low^) are recruited to the tumour via secretion of CSF1 and monocyte chemoattractant protein 1 (MCP1 or CCL2). These monocytes differentiate into M2 polarised macrophages under influence of CSF1, IL4 and IL13 [1,3,19]. The tumour-educated M2-like macrophages resemble trophic/developmental macrophages, downregulate tumouricidal activity and inhibit cytotoxic T cells (CTLs) via IL10 secretion [20]. Instead, they support immunosuppressive, regulatory T cells (Tregs) leading to immune evasion and proliferation of the tumour. M2 polarised TAMs promote an invasive phenotype by supplying promigratory factors as epidermal growth factor (EGF) and they upregulate and secrete cysteine cathepsins in response to IL4 [21,22]. Cysteine cathepsins degrade the extracellular matrix providing a path for invading tumour cells and cleave adherens junction proteins such as E-cadherin [21,23]. Tumour invasion proceeds in lock-step with macrophages, where macrophages lead the migratory front [24,25,26,27]. This macrophage-mediated motility centres around two paracrine signalling loops, comprised of TAM-expressed EGF and tumour-derived CSF1 [28,29]. Interestingly, inhibition of either the EGF or CSF1 signalling pathways results in a constrained lesion in mouse models [26,28].

Tumour expansion is followed by neovascularisation, the key step in the benign-to-malignant transition [2]. Without this ‘angiogenic switch’ the tumour lesion remains severely restricted (2–3 mm^3^) and does not metastasise [10]. In response to the increased hypoxia, TAMs secrete angiogenic factors such as adrenomedullin, VEGF-A and VEGF-C, and produce proangiogenic cytokines as TNFα, IL1β and urokinase-type plasminogen activator (uPA). Moreover, TAM-derived cathepsins and matrix metalloproteases (MMPs) facilitate angiogenesis by releasing growth factors from the ECM and by cleaving ECM components facilitating recruitment of endothelial cells [30,31,32]. Apart from these wound-healing M2 macrophages, there is a specialised subset exclusively involved in neo-angiogenesis, TIE2-expressing macrophages. These TIE2-expressing cells are not recruited with CCL2, but via other tumour-derived chemokines as TIE2-ligand angiopoietin 2, CCL3, CCL5 and CCL8 [32]. These specialised cells are regarded as the major culprit in tumour vascularisation and elimination of this cell type in vivo severely limits the angiogenic switch [16,33].

The last stage in tumourigenesis is dissemination from the primary site. This process is highly inefficient as most circulating malignant cells are cleared by the immune system [34]. TAM-derived proteases support intravasation by degrading the epithelial basement membrane and cleave cell adhesion proteins [15,23]. Following the invasion of blood or lymphatic vasculature, a specialised TAM subset primes the metastatic site by deposition of fibronectin and release of VEGFs and angiopoietins [35]. Furthermore, this pre-metastatic niche provides a platform of chemokines (IL1, IL6 and TNFα), growth factors and adhesion molecules, enabling circulating tumour cells to engraft the secondary site [36].

## 3. Targeting Tumour-Associated Macrophages in the Tumour Microenvironment

The vast majority of TAMs promote tumourigenesis and correspondingly specific depletion of macrophages has shown promising results with in vivo models [37]. Especially macrophage-targeted delivery of liposomal clodronic acid (dichloromethylene diphosphonate) has been extensively researched in preclinical models [2,36,38,39]. Positive effects include severely restricted angiogenesis, disrupted metastasis and a reduced tumour growth. 

Tumour infiltration of TAMs is primarily mediated by CSF1, and genetic ablation of CSF1 in mouse models severely limits tumour progression in comparison with wild type mice [40,41]. Transient disruption of the CSF1-pathway with antibody treatment or antisense oligonucleotides generates a potent antitumour response in preclinical models [42,43]. Moreover, CSF1 is an important regulator of TAM polarisation into an M2 phenotype [11]. The other key mediator in macrophage recruitment is CCL2 (MCP1) and disruption of this cytokine severely limits vascularisation and metastasis in animal models [44,45]. Several clinical trials targeting the CSF1 and CCL2 axes have recently been reviewed by Quail and Joyce [1]. 

The aforementioned strategies focus on the depletion of macrophages from the TME, however, this overlooks the beneficial effects of M1 polarised TAMs. For instance, M1 infiltration is associated with a good prognosis in non-small cell lung cancer (NSCLC) [10]. The tumour-promoting functions of TAM subsets is primarily supported through cytokine crosstalk with the oncogenic microenvironment [3]. Through hijacking this macrophage polarisation and shifting the balance towards an M1-like phenotype in situ, an antitumour immune response can be provoked. This re-education of TAMs has been coined about a decade ago with genetic disruption of nuclear factor kappa beta (NFκB) signalling to shift macrophage polarisation to an M1-like state [46]. The skewed M1 subset was characterised by an increase in iNOS and IL12, and a decreased IL10 and arginase 1 (ARG1) expression. The obtained immunogenic TAMs induced a strong NK cell infiltration in vivo and a sustained tumouricidal effect was observed [46]. Over the last years several small molecules and nanoparticle formulations have been reported to skew macrophage polarisation towards an M1-like state. An overview of these molecules will be presented in the following paragraphs, clustered by molecule type and mechanism of action.

### 3.1. TLR Agonists

The most straightforward method of inducing M1 polarisation is through stimulation with TLR ligands. This activates gene regulators such as NFκβ P50-P65 and signal transducer and activator of transcription 1 (STAT1) (Figure 2). Additionally, TLR4 agonists stimulate activator protein 1 (AP-1) and interferon regulatory factor (IRF) 3 signalling, which leads to a broad activation of M1-associated genes.

For example, Yang et al. described that TLR4 engagement by *Pseudomonas aeruginosa* mannose-sensitive hemagglutinin (PA-MSHA) potently repolarised CD163^+^ TAMs into an M1-like phenotype in vitro, characterised by an increase in TNFα and iNOS expression and a decrease in IL10 and ARG1 [47]. Wang and co-workers found that PA-MSHA re-educated TAMs in a xenograft mouse model of gastric carcinoma. PA-MSHA treatment significantly increased NFκβ signalling, which upregulated IL12 and TNFα expression. Moreover, they found an increase in IFNγ suggesting profound T_h_1 cell infiltration. This treatment inhibited the proliferation and invasion of gastric cancer cells in vitro and demonstrated tumour growth inhibition in vivo [48]. Similarly, Liu and Duan reported that PA-MSHA increased M1 cell infiltration in a mouse model of bladder cancer and inhibited tumour growth [49]. Combination of PA-MSHA with EGF receptor antagonist gefitinib demonstrated a synergistic effect in a xenograft mouse model of NSCLC. Interestingly, PA-MSHA treatment could even resensitise gefitinib-resistant cells both in vitro and in vivo [50]. PA-MSHA is currently in clinical use for treatment of malignant pleural effusion (MPE) in advanced lung cancer patients [47].

Likewise, TLR4 agonism by heat-treated *Mycobacterium indicus pranii* (Mw) was investigated in combination with DTA-1, an agonistic antibody for glucocorticoid-induced TNFR-related protein (GITR). GITR acts as a costimulatory molecule in cytotoxic T cell activation, stimulates IFNγ and IL2 production and suppresses immunosuppressive Treg expansion. Combination of Mw and DTA-1 potently repolarised TAMs in a mouse model of advanced stage melanoma. TAM repolarisation into an M1-like state was confirmed by a significant increase in IL12, iNOS and MHC II, and a decrease in IL10 and TGFβ. DTA-1/Mw combination therapy generated an efficient host immune response and led to almost complete regression of advanced stage solid tumours in mouse models [51].

TLR4 activation has also been investigated with cationic polymers. The positively charged polyethyleneimine (PEI) and cationic dextran (C-dextran) polymers were able to reprogram TAMs towards an M1 phenotype. Reprogramming was associated with an increased IL12, iNOS and MHC II expression and a decreased IL10, ARG1, VEGF and MMP9 expression. Treatment with PEI/C-dextran particles also demonstrated profound CD8^+^ T cell and NK cell infiltration and inhibited cancer growth in an allograft sarcoma model [52].

Vidyarthi et al. recently reported that TLR3 ligand poly I:C reprogrammed macrophages in a mouse model of colon cancer. They observed an increased production of IL6, IL12, TNFα, iNOS, an enhanced antigen uptake and an improved T cell priming. Furthermore, they showed that poly I:C could compete with IL4 and IL13 priming, effectively preventing M2 polarisation. Lastly, they showed that poly I:C reverted M2 TAMs to an M1-like phenotype in a IFNα/β-dependent manner and poly I:C potently inhibited tumour growth in vivo [53].

TLR7 activation is another attractive target for TAM repolarisation as it activates both NFκβ- and IRF7-signalling in macrophages. IRF7 stimulates IFNα production, which leads to a type 1 interferon response. Let-7b is a synthetic miRNA mimic (5′-UGAGGUAGUAGGUUGUGUGGUU-3′) that engages TLR7 and redirects TAMs to an inflammatory phenotype. Furthermore, Let-7b directly suppresses IL10 production in CD4^+^ T cells preventing M2 polarisation [54]. Mannose receptor, C type 1 (MRC1 or CD206)-targeted nanoparticles, loaded with Let-7b, displayed potent TAM repolarisation in vivo, characterised by an increased expression of CD86, MHC II, IL12 and iNOS and a decrease in ARG1 and IL10. Furthermore, Let-7b nanoparticles induced massive infiltration of CD8^+^ T cells and reduced primary tumour volume by up to 70% in a mouse model of breast cancer [55].

Resiquimod (R848) is a synthetic TLR7/8 agonist and was recently reported to potently re-educate TAMs [56]. Rodell and co-workers performed a morphometric mini-screen of proposed reprogramming molecules and noted that especially TLR7 and TLR8 engagement showed potent re-education of macrophages in vitro. Next, they developed a R848-encapsulated TAM-targeted nanoparticle by crosslinking β-cyclodextrin. These nanoparticles effectively accumulated in TAMs in the tumour microenvironment, repolarised them towards an inflammatory phenotype and showed profound tumouricidal effects in a MC38 mouse model of colon cancer. No systemic adverse effects were noted, whereas non-targeted TLR7 or TLR8 agonists have previously shown dose-limiting adverse effects in clinical studies [57,58]. Lastly, it was shown that combination of the R848-nanoparticles with an anti-PD1 checkpoint inhibitor synergised to give improved immunotherapy response and complete rejection of tumours in two out of seven mice. Moreover, mice cured in the course of treatment resisted secondary tumour challenge, indicating that an effective anti-tumour memory had been established [56].

Motolimod is a potent and selective small molecule agonist of TLR8 and is in clinical trials for the treatment of multiple cancers. Motolimod shows potent immune activation even in late-stage cancer patients [59]. In a phase I clinical study it was found that combination of motolimod with EGFR inhibitor cetuximab could potently re-educate TAMs in squamous cell carcinoma patients. Repolarisation was characterised by an increase in IL6, MIP-1β, and MCP1 secretion and an increase in NK cell circulation and activation. Two out of eleven patients displayed partial response and five out of eleven patients showed disease stabilisation. Motolimod is currently under further investigation as adjuvant therapy in a phase II trial as therapy for squamous cell carcinoma [60]. 

TLR9 activation with synthetic unmethylated cytosine-guanine oligodeoxynucleotides (CpG-ODN) leads to activation of interleukin 1 receptor-associated kinase 1 (IRAK)/TNF receptor associated factor (TRAF)-pathway which in turn activates NFκβ and AP-1 signalling [61]. Chen et al. demonstrated that self-assembled CpG-ODN gold nanoparticles could potently stimulate macrophages in vitro and in preclinical models in a TLR9-dependent manner [62]. Guiducci et al. reported that combination of CpG-ODN with an anti-IL10 receptor (anti-IL10R) antibody promptly switched M2 cells to an M1 phenotype and this immunotherapy synergised with adenoviral delivery of CCL16-encoding DNA that increases immune cell infiltration. This three-pronged immunotherapy resulted in remarkable rejection of large tumours in three different allograft mouse models (TSA and 4T1 breast cancers and MC38 colon cancer). The authors described massive haemorrhagic necrosis initiated by activated M1 macrophages and subsequent dendritic cell (DC) and CTL activation leading to almost complete tumour rejection in all three mouse models [63]. Similarly, CpG-ODN was also investigated in combination with an agonistic CD40 antibody in established tumours of B16 melanoma and 9464D neuroblastoma mouse models. The combined immunotherapy potently induced M1 markers (CD40, CD80, CD86, MHC II, TNFα and IL12) in vivo and synergised with chemotherapeutic regimens potently inhibiting tumour growth in melanoma and neuroblastoma mouse models [64].

Lastly, Song et al. investigated manganese dioxide nanoparticles (MnO_2_NP) modified with mannan (Man) and hyaluronic acid (HA) to alleviate tumour hypoxia and reprogram M2 TAMs in a preclinical study. Mannan is a ligand for MRC1 and can be exploited to target M2 macrophages. HA is an important component of the extracellular matrix and low molecular weight HA has been reported to promote M1 polarisation [65]. The HA-coated nanoparticles successfully repolarised TAMs in vivo, which was suggested to be dependent on TLR2 or TLR4 signalling. Finally, combination of Man-HA-MnO_2_NP with chemotherapy significantly inhibited tumour growth and invasion in a mouse model of breast cancer [66].

Altogether, this shows that TLR-mediated TAM repolarisation is a potent antitumour strategy. To date, a limited number of small molecule TLR agonists have been investigated for TAM reprogramming capacity. Several alternative (small molecule) TLR ligands have been synthesised which deserve attention in this context. A few examples have been depicted in Figure 3. [67,68,69,70]. Furthermore, combination of TLR ligands into multivalent conjugates could lead to synergistic effects and phenotypically distinct polarisation states [71,72].

An important consideration of TLR agonism is off-target inflammation, which severely limits the therapeutic window. One way to circumvent systemic inflammation is by selective delivery of the TLR agonist. For instance, Rodell and co-workers showed that encapsulation of TLR ligand R848 in β-cyclodextrin nanoparticles resulted in efficient delivery to TAMs, which potentiated their repolarisation and could prevent systemic inflammation in preclinical models [56]. A second way to increase targeting to TAMs could be by conjugating TLR ligands to antibodies directed to antigens which are overexpressed in TAMs or the TME. For instance, conjugating anti-PD-L1, anti-CSF1R, anti-MRC1 or anti-CD40 antibodies with small molecule TLR ligands would simultaneously target the TME while potentiating the immune response. Ignacio and co-workers recently reviewed several ligation strategies for TLR ligands [73].

### 3.2. Cytokines

A second way of stimulating M1 polarisation is by directly administering proinflammatory cytokines. Similar to TLR agonists, treatment with cytokines is associated with systemic adverse effects [74]. In order to avoid these side effects, intra-tumoural injection or targeted delivery systems have been adopted to selectively target the tumour lesion.

CSF2 is a cytokine that regulates macrophage function by enhancing antigen presentation and immune responsiveness. Engagement of the CSF2 receptor is involved in regulating JAK2/STAT3/5, MAPK, NFκB, and PI3K signalling. Eubank et al. demonstrated that intra-tumoural injection of CSF2 increased the number of TAMs in a mouse model of breast cancer and triggered the ability of TAMs to produce soluble VEGF receptor 1 (sVEGFR1) [75]. sVEGFR1 secretion by macrophages blocks angiogenesis by binding VEGF. Furthermore, CSF2 treatment repolarised M2-like macrophages towards an M1 phenotype, characterised by increased iNOS expression and decreased IL4 and IL10 production. This prevented metastasis and delayed tumour growth in a mouse model of lung cancer [75]. In a similar preclinical study, CSF2 was investigated in combination with 4-iodo-6-phenyl-pyrimidine (4-IPP) an inhibitor of macrophage migration inhibitory factor (MIF). MIF is thought to control M2 polarisation of TAMs in advanced multiple melanoma. Combination treatment demonstrated improved upregulation of M1 markers and downregulation of M2 markers. Moreover, combination therapy increased the tumouricidal activity of repolarised TAMs in a xenograft mouse model of multiple myeloma [76]. 

IL12 is a cytokine produced by macrophages and DCs that promotes CTL expansion. IL12 has been shown to activate multiple signalling pathways including JAK2/STAT4 activation. Moreover, IL12 treatments leads to IFNγ production by T cells and NK cells. To deliver IL12 to the TME several approaches have been investigated. Watkins et al. reported that IL12 containing microspheres were able to reprogram TAMs in a Lewis lung carcinoma (LLC) mouse model, which was associated with a decrease in IL10, MCP1, MIF and TGFβ production and an increase in TNFα, IL15 and IL18 secretion [77]. Wang and co-workers recently reported a pH-responsive nanoparticle designed to deliver IL12 to the tumour lesion. The IL12 nanoparticles potently re-educated TAMs towards a proinflammatory state in a xenograft mouse model of malignant melanoma and resulted in potent immune activation at the tumour site without systemic toxicity [74]. 

Cardoso et al. reported on chitosan/poly (γ-glutamic acid) polyelectrolyte multilayer films (Ch/γ-PGA PEMs) for the delivery of IFNγ to human anti-inflammatory macrophages in vitro. Ch/γ-PGA PEMs caused slow and persistent release of IFNγ and efficiently reprogrammed IL10-polarised macrophages towards an M1-like state, characterised by an increased IL6 production and a decrease in IL10 secretion. Moreover, reprogrammed macrophages adopted M1-like morphology, decreased human gastric cancer cell invasion in vitro and reduced gastric cancer cell-induced angiogenesis in ovo [78].

Lastly, tumour necrosis factor-related apoptosis-inducing ligand (TRAIL) was also reported to skew TAM subsets in vitro. Like other TNF superfamily members TRAIL can induce apoptosis in tumour cells, whereas the majority of non-malignant cells are resistant. Moreover, TRAIL incudes macrophage polarisation through Erk1/2 and NFκβ signalling. In human macrophages co-cultured with human NSCLC cells TRAIL treatment upregulated proinflammatory cytokines such as IL1β, IL6 and TNFα and induced tumouricidal effects in vitro [79]. 

### 3.3. Antibodies

An alternative approach to skew TAM polarisation is to impede anti-inflammatory signalling. Antagonistic antibodies can directly shutdown a specific M2-related pathway, which in turn allows macrophages to adopt a proinflammatory phenotype. For example, Zhu et al. reported on the TAM repolarising effect of an anti-CSF1 antibody in preclinical models. CSF1 blockade decreased M2 tumour infiltration in vivo and reprogrammed resident TAMs to an inflammatory phenotype. This anti-CSF1 treatment sensitised resistant pancreatic ductal adenocarcinoma (PDAC) towards treatment with checkpoint inhibitors (anti-PD1 or anti-cytotoxic T-lymphocyte-associated protein 4 (anti-CTLA4)). The combined immunotherapy led to potent reduction of tumour progression in mouse melanoma and PDAC models, and complete regression of well-established primary tumours in 80% of cases [81]. 

Likewise, combination of an anti-MARCO with an anti-CTLA4 antibody was investigated in mouse model of melanoma. MARCO is a pattern recognition receptor for low-density lipoproteins and is associated with M2 polarisation of macrophages. Anti-MARCO treatment potently repolarised TAMs in an FcγRIIb-dependent manner and in combination with anti-CTLA4 immunotherapy showed enhanced anti-tumour effects in mouse models of melanoma and colon carcinoma [83]. 

Chen and co-workers recently reported on an antagonistic antibody for leukocyte immunoglobulin-like receptor B (LILRB). LILRB family members are negative regulators of myeloid cell activation and correspondingly LILRB blockade increased Erk1/2 and NFκβ signalling and reduced activation of Akt/STAT6 signalling cascades, indicating a shift toward M1 polarised macrophages. Moreover, anti-LILRB2 prevented M2 polarisation by CSF1 or IL4 treatment. Lastly, LILRB2 antagonism potently enhanced the efficacy of anti-PD1 treatment in a mouse model for NSCLC [82]. 

Phosphatidylserine (PS) is a phospholipid that inhibits inflammatory responses. In healthy tissues, PS resides in the plasma membrane and is exposed only upon apoptosis. During apoptosis, PS receptors (PSR) on macrophages and DCs bind exposed PS on dying cells, which promotes tolerance by triggering IL10 and TGFβ expression. Endothelial cells in the tumour vasculature and tumor-derived microvesicles constitutively expose PS, hereby suppressing immune cell function [131]. Correspondingly, a PS-binding antibody, 2aG4, re-educated M2 macrophages and increased DC cell maturation. This resulted in profound destruction of tumour vasculature and reduction in tumour size in vivo. Combination of anti-PS with docetaxel synergised to reject established PC3 and LNCaP prostate cancers in mice [84]. Similarly, Cheng and co-workers reported on a combination therapy of 2aG4 with tyrosine kinase inhibitor sorafenib in a mouse model of hepatocellular carcinoma (HCC) and performed a small phase I clinical trial. They found that combination therapy synergised to increase the number of apoptotic vessels and displayed stronger tumour growth inhibition as compared to single agent therapy in mouse models. Next, they reported that combination treatment with a humanised, monoclonal PS antibody, bavituximab, was well tolerated in HCC patients. A phase II clinical trial with bavituximab in HCC is currently ongoing [86]. Chalasani et al. reported on bavituximab as adjuvant therapy to paclitaxel in a phase I clinical trial for HER2-negative breast cancer. Treatment was well-tolerated and resulted in an overall response of 85% [85]. Lastly, bavituximab has been investigated in a phase III clinical trial in advanced stage lung cancer. Sadly, the combination of bavituximab with docetaxel was not superior to docetaxel alone in patients previously treated NSCLC and the addition of bavituximab did not significantly change systemic adverse effects. However, in patients treated with immune checkpoint inhibitors post-study, the bavituximab-treated cohorts displayed an increased overall survival as compared to placebo-treated patients. As the authors mentioned, this would require further investigation, though, one could hypothesise a contribution from TAM repolarisation [87]. 

The hypoxic milieu of the TME is also an important driver of M2 polarisation. Hypoxia leads to the activation of hypoxia-inducible factors (HIFs), which increases the expression of M2 associated proteins as ARG1, VEGF, MMP7, MMP9 and COX2 [5]. Hypoxia also increases the expression of triggering receptor expressed on myeloid cells 1 (TREM1) on both M1 and M2 macrophages. TREM1 is a member of the immunoglobulin-like receptor family and has a critical role in acute inflammation. Activation of TREM1 induces iNOS, IL1β, IL6 and TNFα expression and hence drives M1 polarisation [132]. Accordingly, Raggi and co-workers demonstrated that an agonistic TREM1 antibody was able to reverse hypoxia-induced M2 polarisation towards an inflammatory phenotype in vitro. This reprogramming was illustrated by an increase in IL6, IL12 and TNFα [88]. Huang et al. sought to alleviate the hypoxic environment by employing an anti-VEGFR2 antibody (DC101). They reported that low-dose, not high-dose, anti-VEGFR2 treatment leads to improved vessel perfusion and subsequent repolarisation of TAMs towards an M1 phenotype, characterised by an increase in iNOS, IL12, CXCL9 expression. Furthermore, combination of vessel-normalising doses of anti-VEGFR2 treatment synergised with cancer vaccination in the treatment of orthotopic breast cancer in mouse models [89]. Similarly, a bispecific, antagonistic antibody (A2V) directed against VEGF and angiopoietin-2 (Ang2) demonstrated M2 repolarisation in vivo. A2V treatment reduced MRC1 expression and increased CXCL9 production in TAMs, though a more detailed analysis of TAM polarisation was not reported. The bispecific antibody, A2V, was able to enhance vessel pruning, reduce tumour burden and prolong survival in Gl261 and MGG8 glioblastoma models [90]. 

### 3.4. RNAs

A second method of inhibiting M2-related signalling pathways is by employing antisense microRNAs (miRNAs). miRNAs are small noncoding RNA fragments that play an important role in posttranscriptional gene suppression. For instance, miRNA-155 and miRNA-125b are important regulators of macrophage polarisation and present interesting therapeutic targets. miRNA-155 is upregulated in an NFκβ-mediated manner, directly enhances TNFα translation and suppresses anti-inflammatory pathways as suppressor of cytokine signalling 1 (SOCS1) and inositol polyphosphate-5-phosphatase (INPP5D) signalling [133]. Likewise, miRNA-125b is also enriched in M1 polarised macrophages and suppresses IRF4, a negative regulator of IRF5 signalling [91]. miRNAs are also important in cell-cell signalling and can be transported via exosomal vesicles. Su et al. reported on transfection of human PDAC cells with miRNA-125b and miRNA-155 HA-nanoparticles. The transfected tumour cells upregulated miRNA-125b and miRNA-155 expression in exosomes and repolarised mouse M2 macrophages in a transwell co-culture assay. Parayath and co-workers recently showed that transfection of peritoneal macrophages with miRNA-125b-containing HA-nanoparticles repolarised macrophages in a spontaneous NSCLC mouse model, illustrated by a 300-fold increase of iNOS/ARG1 ratio [92]. Similarly, Ortega et al. reported on the development of a MRC1-targeted nanoparticle for the delivery of siRNA against nuclear factor of kappa light polypeptide gene enhancer in B-cells inhibitor alpha (IκBα). IκBα is a suppressor of NFκβ activation and accordingly IκBα-knockdown showed an increase in TNFα and CXCl9 and a decrease in IL10 and MRC1 secretion, effectively re-educating macrophages into an M1-like state in vitro [93].

Alternatively, Seif and co-workers looked at the delivery of MyD88- and TNFα-encoding mRNA to induce M1 polarisation. They employed opsonised recombinant *Saccharomyces cerevisiae* as a novel delivery vehicle. This empty vehicle alone induced TLR-engagement as illustrated by the repolarisation of M2 macrophages in vitro. The encapsulation of MyD88 and TNFα RNAs prolonged this activation and successfully reprogrammed M2 macrophages to an antitumour M1 phenotype [94]. 

### 3.5. Small Molecules

Over recent years, several small molecules have been identified that demonstrate effects on TAM polarisation, such as cyclooxygenase 2 (COX2) inhibitors, tyrosine kinase inhibitors (TKIs), histone deacetylase (HDAC) inhibitors and stimulator of interferon genes (STING) agonists (Figure 4), additional to the aforementioned small molecule TLR ligands (Section 3.1). With the identification of these compounds several molecular mechanisms involved in TAM polarisation have been unravelled, leading to the identification of potential therapeutic targets. 

Several COX2 inhibitors have been identified to shift macrophages into an inflammatory phenotype [95,96,97]. COX2 inhibition disrupts prostaglandin 2 (PGE2) signalling, which in turn increases pro-inflammatory signals as TNFα, IL12, and iNOS production in macrophages. Chronic inhibition of COX2 with inhibitor NS-398 was reported to increase TNFα and IL12 production when challenged with LPS in vivo [95]. Likewise, COX2 selective inhibitor etodolac induced an M1 phenotype in a syngeneic 4T1 breast cancer model. M1 polarisation was characterised by an increase in MHC II, CD80 and CD86 expression and reduced IL10, TGFβ, MMP1, MMP9, VEGF-A and VEGF-C secretion. Etodolac treatment inhibited lung metastasis formation, but did not affect primary tumour size [96]. In a preclinical model of colorectal cancer, selective COX2-inhibitor celecoxib induced an inflammatory M1-like phenotype, reduced metastatic behaviour and primary tumour burden [97]. Lastly, combination of celecoxib with EGFR inhibitor gefitinib was reported in a phase I clinical for the treatment of docetaxel-resistant prostate cancer. Combination treatment was well tolerated and inhibited tumour growth and tumour invasion in patients [134]. 

Downey et al. reported on two STING agonists, 5,6-dimethylxanthenone-4-acetic acid (DMXAA) and cyclic guanosine monophosphate–adenosine monophosphate (2′3′-cGAMP) that were able to repolarise macrophages in mouse models for NSCLC and breast cancer. Moreover, DMXAA is a well-known vasculature disrupting agent. Both compounds showed TAM repolarisation in vitro, characterised by an increase in iNOS, IFNγ and IL12, and a decrease of MRC1. DMXAA-treatment in mouse models lead to haemorrhagic necrosis in subcutaneous lung and breast tumours, but could not show the same effects in metastatic sites. The authors reasoned that his was due the difference in vascular structures between primary and metastatic tumours [98]. 

CSF1R-inhibitors have also been reported to repolarise TAMs. Though the CSF1/CSF1R axis is primarily associated with macrophage recruitment, CSF1R engagement also plays an important role in macrophage polarisation. Ao and co-workers found that pexidartinib did not decrease TAM infiltration but reprogrammed TAMs in a xenograft model of hepatocellular cancer. Pexidartinib increased CSF2 and IFNγ in the TME and displayed increased CTL infiltration [99]. Yan et al. reported that CSF1R-inhibitor pexidartinib could potently re-educate macrophages in a mouse model for glioma while maintaining macrophage infiltration. Repolarisation was characterised by a downregulation of ARG1, MRC1, CD163 and IL10, though no upregulation of M1 markers was noted. Pexidartinib potently inhibited tumour growth and combination of pexidartinib with TKI dovitinib or vatalanib synergised to reject existing tumours in mouse models [100]. Similarly, treatment of glioblastoma with CSF1R inhibitor BLZ945 reprogrammed TAMs to an M1-like state. BLZ945 downregulated ARG1, MRC1 and adrenomedullin, and upregulated IL-1β expression. Lastly, CSF1R inhibition significantly reduced established tumours and increased disease-free survival in mouse model for glioblastoma [101].

Another way to prevent M2 polarisation is inhibition of Bruton’s tyrosine kinase (BTK). BTK plays an important role in B cell-macrophage interactions, drives M2 polarisation and immune suppression in PDAC. BTK signalling is dependent on phosphoinositide 3-kinase-γ (PI3Kγ). Correspondingly, inhibition of BTK with ibrutinib or inhibition of PI3Kγ with TG100-115 skewed TAMs towards an inflammatory phenotype as characterised by an increased expression of IL12 and a decreased expression of TFGβ and ARG1 in vitro. Moreover, inhibition of BTK or PI3Kγ signalling suppressed tumour growth in a mouse model for PDAC [102]. Ibrutinib is currently approved for the treatment of mantle cell lymphoma, chronic lymphocytic leukemia, and Waldenström’s macroglobulinemia. 

HS-1793, a synthetic resveratrol analogue, was reported to decrease CD206^+^ TAM infiltration, downregulate TGFβ production, and upregulate IFNγ production in tumour bearing mice. TAM reprogramming was suggested to be in an IFNγ-dependent manner [103]. Secondly, Kim et al. recently reported that HS-1793 inhibited HIF1α activity, suppressed VEGF secretion and inhibited tumour growth in a mouse model of breast cancer [104].

Vorinostat, a HDAC inhibitor, was recently reported as adjuvant therapy to skew macrophages into an M1 state. Peng et al. developed a dual-targeted co-delivery system for a combination of gefitinib and vorinostat. The trastuzumab-coated, mannosylated liposomal system efficiently targeted both HER2-positive cancer cells and MRC1-positive TAMs, and reprogrammed protumoural, M2 polarised TAMs to an M1 phenotype, in an HDAC2-dependent manner. TAM repolarisation was illustrated by decreased ARG1, IL10 and CD206 levels and increased iNOS, CD86, TNFα and ROS levels. This treatment re-sensitised gefitinib resistant H1975 lung cancer and potently inhibited tumour growth in mouse models [106]. Similarly, a combination of vorinostat and TKI sorafenib was recently reported to increase infiltration of F4/80^+^ iNOS^+^ M1 macrophages in a mouse model for PDAC. Combination treatment showed synergy with anti-PD1 treatment which demonstrated a further reduction in tumour growth in comparison with anti-PD1 treatment alone. A phase I clinical trial of the vorinostat and sorafenib combination is currently ongoing in hepatocellular carcinoma patients [107].

Sorafinib alone has also been shown to influence macrophage polarisation in vitro. Sorafinib-treatment potentiated M0 macrophages to LPS-mediated polarisation, which was characterised by a profound increase in IL6, IL12 and TNFα production as compared to LPS alone. Secondly, sorafinib altered NK cell and TAM crosstalk, which lead to upregulated IFNγ expression in NK cells and increased NK cell migratory activity in vitro [108]. Alsaab et al. recently reported on combination of sorafenib with apoptosis inducer CFM4.16 in a hypoxia-targeted nanoparticle. This nanoparticle delivery subsequently reprogrammed macrophages to an M1-like state in vitro, via inhibition of AKT1 phosphorylation preventing the activation of STAT6. Nanoparticle-treatment potently inhibited tumour growth in a mouse model of renal cell carcinoma [109]. Similarly, Zhang et al. recently reported on the combination of sorafinib with HDAC inhibitor panobinostat and bromodomain protein inhibitor OTX015. This triple combination led to effective rejection of tumours in a U87-EGFRvIII glioblastoma model and a xenograft mouse model with patient-derived glioblastoma cells [110]. Sorafenib is currently approved for primary kidney cancer, advanced primary liver cancer, and radioactive iodine resistant advanced thyroid carcinoma.

TKI sunitinib was reported to skew macrophages when used in combination with an agonistic antibody against GITR. Interestingly, combination treatment potently increased MHC II, CD80, CD86, CXCL10 and IL12 in TAMs but not with either treatment alone. The combination treatment also increased T cell and NK infiltration, and upregulated IFNγ production in a model for metastatic renal cell carcinoma (RCC) [111]. 

Tan and co-workers reported that baicalin, a natural flavonoid found in several medicinal plants, was able to reprogram TAMs in an orthotropic mouse model of HCC. Baicalin activated the RelB/P52 complex, a regulator of non-canonical NFκβ signalling. Activation of the RelB/P52 pathway was dependent on TRAF2 lysosomal degradation, induced by baicalin-mediated autophagy. Baicalin-treatment of HCC mice significantly upregulated M1 markers as CD86, TNFα, and IL12, and downregulated M2 markers IL10 and ARG1. Baicalin treatment diminished cell proliferation and tumour motility, but could not reduce primary tumour size in HCC mice [112].

Chlorogenic acid (CHA), a phenolic compound found in coffee, apples and green tea, has been investigated in several clinical trials and demonstrated therapeutic effects in colon, lung, brain and breast cancers. Xue et al. studied the mechanism underlying this anti-cancer activity by looking at CHA-mediated reprogramming of IL4 polarised M2 macrophages in vitro. CHA upregulated expression of iNOS via STAT1 induction, while decreasing the expression of M2 markers CD206, ARG1 and IL10 by downregulating STAT6 in M2 macrophages. CHA-treatment repolarised TAMs and displayed partial inhibition of tumour growth in a G422 xenograft mouse model [113]. 

Emodin, a natural product isolated from rhubarb, suppresses STAT6 phosphorylation, which lead to a decrease in MMP9 and survivin activity. In a mouse model of pancreatic cancer emodin-treatment reduced primary tumour volume up to 25 percent, and decreased angiogenesis, invasion and metastasis. [114,115]. 

Similarly, hydrazinocurcumin (HC), a synthetic analogue of natural compound curcumin, reprogrammed TAMs in a STAT3-dependent manner. Zhang et al. developed a legumain-targeted nanoparticle loaded with HC which showed potent repolarisation of TAMs in vivo, halted tumour growth and prolonged survival in a mouse model of 4T1 breast cancer [116].

Ball and co-workers reported on a synthetic analogue of natural compound oleanolic acid, 2-Cyano-3,12-dioxo-oleana-1,9(11)-dien-28-oic acid methyl ester (CDDO-Me), that was able to help skew macrophages to an inflammatory phenotype. Treatment with CDDO-Me before treatment with LPS potentiated TAMs to the effects of LPS-stimulation. This co-stimulation reprogrammed TAMs ex vivo, characterised by an increased expression of TNFα, IFNγ, CXCL9 and IL6 and decreased expression of IL10, ARG1, VEGF and SOCS3. Furthermore, CDDO-Me-treatment of TAMS inhibited endothelial cell tube formation in vitro [117]. Though the exact molecular mechanism underlying this reprogramming is not fully understood, CDDO-Me has been reported to inhibit tumour growth in several in vivo studies. For instance, CDDO-Me treatment lead to potent inhibition of tumour growth and prevented cancer recurrence in xenograft mouse models for BxPC3 pancreatic cancer and MiaPaCa-2-Luc PDAC [135].

Qin et al. demonstrated that dopamine repolarised TAMs and inhibited cancer development following vascular normalisation [106]. Dopamine also directly repolarised TAMs in a dopamine-receptor 2 (DR2) mediated manner, characterised by an increase of iNOS, CXCL9, TNFα and IL12 and decrease in ARG1, VEGF, IL10, MRC1 and TGFβ. The exact molecular mechanism following DR2 activation remains elusive [118].

Copper *N*-(2-hydroxy acetophenone) glycinate (CuNG), a synthetic copper chelate, was reported to reprogram M2 TAMs. Copper plays an important role in macrophage homeostasis and influences numerous signalling pathways. For example, an increased copper concentration in macrophages correlates with the production of inflammatory cytokines. Secondly, CuNG is a redox-active compound and can affect ROS generation. CuNG reprogramming was characterised with elevated levels of IL12 and IFNγ, and decreased levels of IL10 and TGFβ. Consequently, CuNG was able to induce an anti-tumour immune response in an Ehrlich–Lettre ascites carcinoma mouse model [119]. In a follow-up study, the reprogramming capacity of CuNG was demonstrated to be dependent on ROS generation [120].

Zoledronic acid (ZA), a third-generation bisphosphonate, is currently in clinical use to treat bone metastasis. Coscia et al. reported that ZA also shows tumour inhibiting activity to primary tumours in a spontaneous breast cancer model. ZA-treatment upregulated iNOS and IFNγ in the TME and showed a decrease in IL10. TAMs macrophages were reported to have an upregulated NFκβ signalling and an increase in iNOS production. ZA delayed tumour growth and extended survival in ErbB-2 mice [121]. Secondly, ZA has been reported to decrease CCL2 expression in early-stage breast cancer model, preventing TAM recruitment and reducing tumour growth [122]. Lastly, Comito and co-workers reported that ZA-treatment reduced angiogenesis, inhibited tumour invasion and prevented M2 polarisation of macrophages by decreasing IL10 levels in the TME [123]. ZA has been investigated in a multitude of clinical trials and is currently used to treat bone metastasis and osteoporosis.

Likewise, metformin, first-line medication of type 2 diabetes, was reported to prevent M2 polarisation in several cancer models. Metformin activates AMPK1 signalling, an important kinase in the NFκB-pathway, which suppressed IL13 induced polarisation of macrophages [124]. Moreover, metformin-treatment lowered expression of IL4, IL10 and IL13 in tumour cells and inhibited metastasis formation and angiogenesis in a mouse model of LLC, a xenograft model of breast and prostate cancer. Sadly, metformin showed very limited inhibition of primary tumour growth in vivo [124,125,126]. 

Chloroquine (QC), an anti-malarial drug, was recently reported to prevent M2 macrophage polarisation in a 4T1 breast cancer model. QC suppressed TGFβ and IL10 production in the TME and inhibited the recruitment of Tregs and M2 macrophages. QC partially inhibited cancer growth in vivo and prolonged survival in mice [127]. Chen and co-workers reported that QC directly reprogrammed macrophages by stimulating lysosomal calcium release. The weakly basic QC accumulated in acidic lysosomes, leading to an increase in cytoplasmic calcium, which activates P38/NFκβ signalling. This induced expression of IFNγ, IL12, iNOS, CD80, CD86 and MHC II, and decreased expression of ARG1, CD206, and IL10. Lastly, QC inhibited tumour growth of B16 melanoma and H22 hepatocarcinoma mouse models which synergised with cyclophosphamide treatment [128].

### 3.6. Others

Finally, two polymeric molecules have been described to drive macrophages to an M1 phenotype. β-glucan is a natural oligo-saccharide isolated from yeast that has been investigated for tumouricidal activity. Liu et al. reported that β-glucan repolarises macrophages to an M1 phenotype in a dectin-1-dependent fashion. Dectin-1 is a c-type lectin receptor for a variety of β-1,3-linked and β-1,6-linked glucans, and dectin-1 agonism leads to NFκβ activation. Correspondingly, β-glucan upregulated iNOS, IL1β, IL6, IL12 and TNFα and downregulated IL10 and ARG1 in a mouse model of NSCLC. Furthermore, β-glucan reduced tumour growth and inhibited vascularisation [130].

Secondly, Rolny et al. reported on a genetic gain-of-function strategy to study histidine-rich glycoprotein (HRG) in the TME. HRG overexpression was shown to increase IL6 and IL12 production in macrophages as compared to HRG^low^ tumours. This TAM repolarisation was found to be dependent on downregulation of placental growth factor (PGF). PGF is a member of the VEGF family and an important regulator of vascularisation. Lastly, HRG^high^ 4T1 tumours displayed vascular normalisation, delayed growth rate and an impaired metastasis in mice. Treatment with doxorubicin was potentiated by HRG overexpression, most likely due to the increased vessel perfusion [129].

## 4. Conclusion

Myeloid cell infiltration in the TME has been designated as a key factor in the proliferation and dissemination of cancerous growth. TAMs form the major portion of these myeloid cells and are reported to mediate in an array of pro-oncogenic processes. In each of these processes specialised TAM subsets can be distinguished, showing characteristics of both M1 and M2 macrophages, resembling trophic/developmental macrophages in healthy tissues. These distinct phenotypes arise from the highly plastic nature of macrophages and are maintained primarily via the crosstalk with the TME. Overruling these environmental cues is the key factor in successful application of TAM-targeted immunotherapy and requires a detailed understanding of the molecular wheelwork underlying TAM polarisation. 

In the recent years various molecules have been reported to skew TAM subsets into an inflammatory phenotype. This has led to the identification of key cellular pathways involved in macrophage polarisation. Robust anti-metastatic and anti-angiogenic effects are well-documented, however, most of the described molecules lack the potency required to elicit durable tumouricidal effects in preclinical models. The discovery and development of dedicated small molecules will be of substantial value to overcome this challenge. For instance, recent reports have identified that small molecule TLR ligands provide potent repolarisation of TAMs in mouse models [56,59]. Subsequent combination of TAM reprogramming with checkpoint inhibitors (PD1 or CTLA4 antagonists) or stimulating antibodies (CD40 or GITR agonists) displayed a remarkable synergy [51,56,64].

Secondly, TAM repolarisation showed an interesting interplay with the tumour vasculature. On the one hand, re-educated TAMs decrease expression of angiogenic growth factors leading to vessel normalisation. Vice versa, anti-VEGFR or anti-VEGF/anti-angiopoietin treatments decreased vascular abnormality and reduced hypoxia, which led to repolarisation of TAMs. Vessel normalisation was shown to sensitise the tumour to chemotherapeutic regimens (docetaxel, paclitaxel or doxorubicin) and improved CTL and NK cell infiltration. Hence, combination of TAM reprogramming with vascular normalisation strategies might prove an attractive strategy in cancer treatment.

A major challenge of TAM repolarisation are off-target inflammatory responses. To prevent systemic inflammation targeting strategies can be adopted to selectively reach TAMs or the TME. For instance, antibodies directed against upregulated antigens could be ligated to nanoparticle-encapsulated molecules or directly conjugated to small molecule TLR ligands. Lastly, conjugating multiple TLR ligands together could be an attractive strategy to potentiate TAM reprogramming.

## Figures and Tables

**Figure 1 molecules-24-00009-f001:**
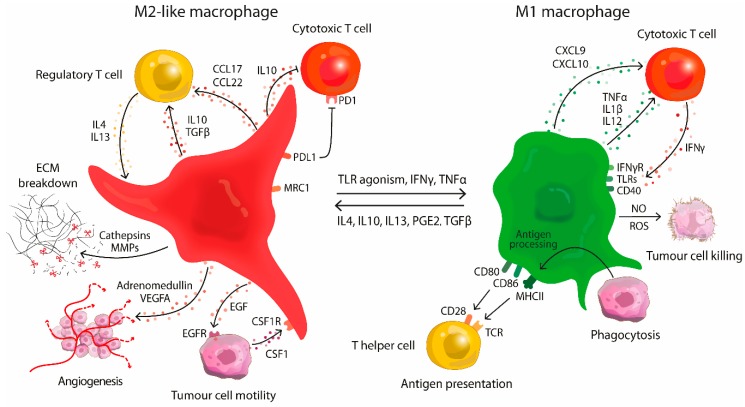
Schematic representation of the functions of M1 and M2-like macrophages in tumour development. During early stages of tumourigenesis activated (M1) macrophages present antigens and support cytotoxic T lymphocytes (CTL) by the production of proinflammatory cytokines. They eliminate tumour cells with nitrogen and oxygen radicals or by phagocytosis. These antitumour macrophages can be seized by the tumour and shifted to an M2-like state by secretion of immunosuppressive cytokines. The formed M2-like macrophages suppress CTL function and redirect them to immunosuppressive T cell subsets. M2 polarised TAMs support tumour growth in all stages of disease including proliferation, angiogenesis and metastasis.

**Figure 2 molecules-24-00009-f002:**
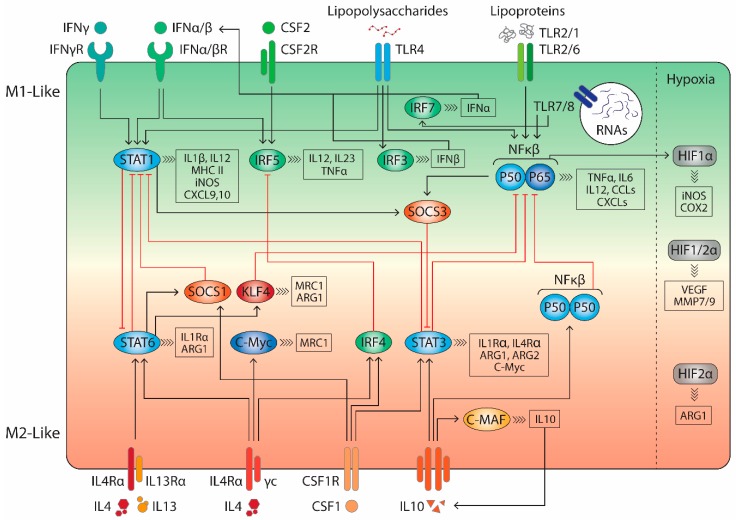
Schematic representation of the key signalling pathways driving macrophage polarisation. M1 stimuli as IFNγ, bacterial lipoproteins or lipopolysaccharides induce IRF3, IRF5, IRF7, STAT1 and P50-P65 NFκβ signalling leading to a proinflammatory response. In contrast, M2 stimuli as IL4, IL10 and IL13 activate IRF4, STAT3, STAT6 and P50-P50 NFκβ signalling resulting in anti-inflammatory gene expression and tumour progression. The crosstalk between these regulatory pathways determines the exact outcome of macrophage activity.

**Figure 3 molecules-24-00009-f003:**
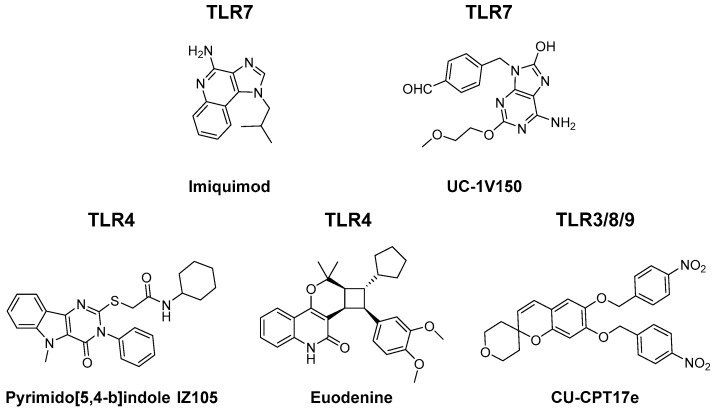
Structure of small molecule TLR agonists with potential TAM reprogramming abilities.

**Figure 4 molecules-24-00009-f004:**
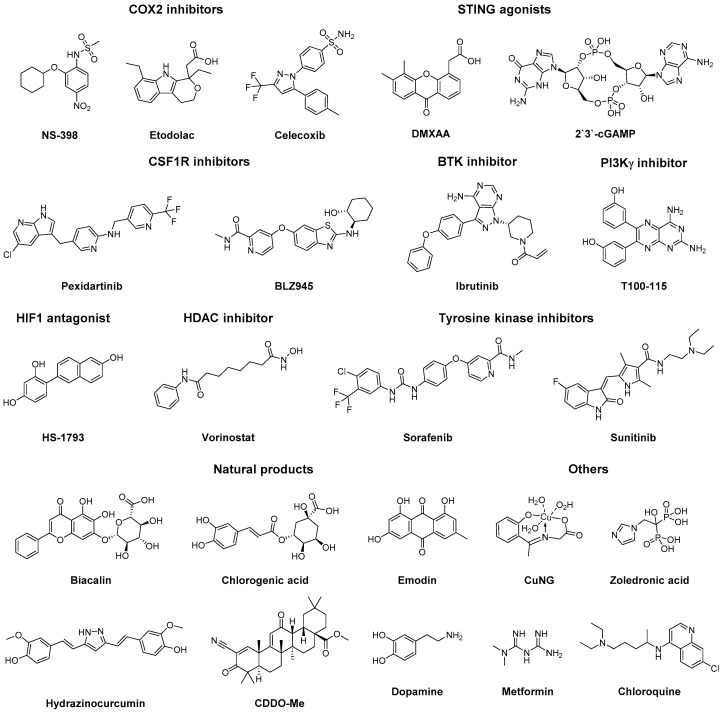
Chemical structure of TAM reprogramming small molecules.

**Table 1 molecules-24-00009-t001:** TAM reprogramming molecules.

Molecule	Target	Signalling Pathway	Type of Study	Reference	Comments
**TLR Agonists**
Poly I:C	TLR3	NFκβ	In vitro and in vivo	[53]	
Cationic polymers PEI/ C-dextran	TLR4	NFκβ and IRF3	In vitro and in vivo	[52]	
Mycobacterium idicus pranii	TLR4	NFκβ and AP-1/MAPK P38	In vivo	[51]	Studied in combination with DTA-1
*pseudomonas aeruginosa* mannose-sensitive hemagglutinin (PA-MSHA)	TLR4	NFκβ and IRF3	In vitro and in vivo	[47,48,49,50]	Approved for advanced lung cancer
Let-7b microRNA mimic	TLR7 and anti-IL10	NFκβ and IRF7	In vivo	[55]	Administered in a MRC1-targeted nanoparticle
Resiquimod (R848)	TLR7/8	NFκβ and IRF7	In vitro and In vivo	[56]	Administered in a β-cyclodextrin nanoparticle
Motolimod	TLR8	NFκβ	In vivo and phase I/II clinical trials	[59,60,80]	
CpG motifs	TLR9	NFκβ and AP-1	In vivo	[62,63,64]	
Hyaluronic acid	TLR-2 or TLR-4	NFκB or IRF3	In vivo	[66]	
**Cytokines**
CSF2	CSF2 receptor	JAK/STAT3/5, MAPK, NFκβ, and PI3K	In vivo	[75,76]	Studied in combination with 4-IPP
IL12	IL12 receptor	JAK2/STAT4	In vivo	[74,77]	Administered in a poly(β-amino ester) nanoparticle
IFNγ	IFNγ receptor	STAT1	In vitro	[78]	Administered as a chitosan/poly(γ-glutamic acid) nanoparticle
TRAIL	TRAIL receptor 1	NFκβ and ERK1/2	In vitro	[79]	
**Antibodies**
Anti-CSF1	CSF1	NFκβ, ERK1/2 and miR21	In vivo	[81]	
Anti-LILRB2	LILRB2 receptor	NFκβ, Erk1/2 and Blocks Akt/STAT6	In vitro and in vivo	[82]	
Anti-MARCO	MARCO	FcγRIIB	In vivo	[83]	
Anti-CD40	CD40	NFκβ, ERK1/2 and P38 MAPK	In vivo	[64]	Studied in combination with CpG-ODN
Anti-IL10 receptor	IL10Rα	Blocks Akt/STAT3	in vivo	[63]	Studied in combination with CpG-ODN and CCL16
Anti-phosphatidylserine	Phosphatidylserine	FcγRII/III	In vivo and phase I/II/III clinical trials	[84,85,86,87]	
Anti-TREM-1	TREM-1	TREM-1/DAP12/Syk	In vitro	[88]	
Anti-VEGFR-2	VEGF receptor 2	Decreased hypoxia sensing	In vivo	[89]	
Bispecific anti-angiopoietin-2 anti-VEGF-A antibody	Angiopoietin-2 and VEGF-A	Decreased hypoxia sensing	In vivo	[90]	
**RNAs**
miR155/miR125b2	TNFα/SOCS1/IRF4	Enhances TNFα translation and blocks SOCS1 and IRF4	In vitro	[91,92]	Administered in a hyaluronic acid-based nanoparticle
IκBα siRNA	IκBα	NFκβ	In vitro	[93]	Administered in a mannosylated nanoparticle
MyD88/TNF mRNA in *S. cerevisae*	MYD88/TNF receptor	NFκβ and AP-1	In vitro	[94]	Empty *S. cerevisae* also activates macrophages
**Small Molecules**
NS-398	COX2 inhibitor	Blocks PI3K/Akt	In vivo	[95]	
Etodolac	COX2 inhibitor	Blocks PI3K/Akt	In vivo	[96]	
Celecoxib	COX2 inhibitor	Blocks PI3K/Akt	In vivo	[97]	
DMXAA	STING-agonist	TBK1/NFκβ	In vivo	[98]	
2‘3‘-cGAMP	STING-agonist	TBK1/NFκβ	In vitro	[98]	
Pexidartinib	CSF1R-inhibitor	STAT3, IRF4	In vivo	[99,100]	
BLZ945	CSF1R-inhibitor	STAT3, IRF4	In vivo	[101]	
Ibrutinib	BTK inhibitor	Blocks BTK	In vitro	[102]	Approved for leukemia
TG100-115	PI3Kγ inhibitor	Blocks PI3Kγ	In vitro	[102]	
HS-1793	HIF1 antagonist	JAK/STAT1	In vitro	[103,104]	
Vorinostat	HDAC inhibitor	HDAC2	In vivo	[105,106,107]	Administered in a redox-responsive nanoparticle
Sorafenib	Tyrosine kinase inhibitor	Blocks Akt/STAT6	In vivo	[108,109,110]	Approved for advanced kidney cancer
Sunitinib	Tyrosine kinase inhibitor	Blocks STAT3	In vivo	[111]	Studied in combination with anti-GITR, approved for renal and GI cancers
Baicalin	Unknown	RelB/P52	In vivo	[112]	
Chlorogenic acid	Unknown	Activates STAT1 and blocks STAT6	in vivo	[113]	
Emodin	Unknown	Blocks Akt/STAT6	In vivo	[114,115]	
Hydrazinocurcumin	Unknown	Blocks STAT3	In vivo	[116]	
CDDO-Me	Unknown	Unknown	In vivo	[117]	
Dopamine	Dopamine receptor 2	Unknown	In vivo	[118]	
CuNG	ROS generation	MAPK P38, ERK1/2, NFκβ and AP-1	In vivo	[119,120]	
Zoledronic acid	Unknown	NFκβ	In vivo	[121,122,123]	Approved for osteoporosis and bone metastases
Metformin	Unknown	AMPK/NFκβ	In vivo	[124,125,126]	
Chloroquine	Unknown	NFκβ, P38 MAPK and TFEB	In vivo	[127,128]	
**Others**
Histidine-rich glycoprotein	PIGF	Unknown	In vivo	[129]	
β-Glucan	Dectin-1	Erk1/2	In vivo	[130]

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
