# Peer review of "Molecular Repolarisation of Tumour-Associated Macrophages"

_molecules, 2018, doi:10.3390/molecules24010009_

Reviewer 1 Report

The manuscript is well written and the citation is updated. However, the information on the origin of TAM is lacking. The authors should describe the origin of TAMs and the molecular mechanisms underlying their polarization into M2 state. Moreover, some clinical studies were described intermingling with studies on mouse models. The authors should describe explicitly the types of studies, clinical or pre-clinical ones, of each citation. Furthermore, the manuscript lacks the description on the molecular and pharmacological mechanisms of the agents, particularly small molecules. The authors should try to discuss the pharmacological mechanisms of small molecules as much as possible.

Author Response

Dear Nancy Li,

Please find enclosed our revised manuscript entitled "Molecular Repolarisation of Tumour-Associated Macrophages" (Manuscript ID: molecules-402551). We were happy to see that both reviewers were very positive about our work and we appreciate the suggestions made to improve our manuscript. We have revised the manuscript in response to the suggestions and comments by the reviewers and we have highlighted all revisions using the "Track Changes" function in Microsoft Word. Please find below our response to the reviewers.

Response to reviewer #1:

1.      However, the information on the origin of TAM is lacking. The authors should describe the origin of TAMs and the molecular mechanisms underlying their polarization into M2 state.

Response: We thank the reviewer for pointing this out. We have further clarified the origin of TAMs in the introductory section (lines 80-89) by slightly rephrasing the original text and the addition of a sentence to clarify where TAMs originate from and the molecular mechanism that determines their polarisation state. This section now reads:

“Bone marrow-derived monocytes (F4/80low) are recruited to the tumour via secretion of CSF1 and monocyte chemoattractant protein 1 (MCP1 or CCL2). These monocytes differentiate into M2-polarised macrophages under influence of CSF1, IL4 and IL13 [1,3,19].”

Several reviews have covered this subject and are referenced in the text.

2.      Moreover, some clinical studies were described intermingling with studies on mouse models. The authors should describe explicitly the types of studies, clinical or pre-clinical ones, of each citation.

Response: We thank both reviewers for this suggestion. We have reviewed and updated the text to state more clearly the type of study. Adaptations are indicated by the “track changes” function in Microsoft Word.

3.      Furthermore, the manuscript lacks the description on the molecular and pharmacological mechanisms of the agents, particularly small molecules. The authors should try to discuss the pharmacological mechanisms of small molecules as much as possible.

Response: We fully agree that the pharmacological mechanisms are a focal point of the manuscript and we endeavoured to review all the available information on the molecular targets and polarisation mechanisms of the presented TAM repolarisation molecules. Sadly, for some of the presented molecules, for instance baicalin, chlorogenic acid, emodin, hydrazinecurcumin etc., the molecular target has not yet been elucidated. In these cases, the mechanism of action that has been hypothesised by the original authors has been reviewed in the text. An overview of the polarisation pathways are presented in table 1. When the target and the signalling pathway are not known this has been stated in Table 1. We have reviewed and described all available information to the best of our knowledge.

Response to reviewer #2:

1.      It would be helpful if the authors could state clearly whether the described experimental results (or conclusions) were derived from mouse or human studies, so that the readers can be aware of the real-life implications of these results.

Response: As stated above in response to reviewer 1, we thank both reviewers for their suggestion and we have reviewed and updated the text to state more clearly the type of study. Adaptations are indicated by the “track changes” function in Microsoft Word.

2.      There are several statements without supporting references. For instance, in Line 33, Page 1, the statement “TAMs are responsible for…from attacking the tumour.” needs a reference. In Line 34, Page 1, the statement “Instead, TAMs enhance tumour development…with a poor prognosis.” needs a reference. In Line 98, Page 3, “Tumour expansion is followed by…benign-to-malignant transition.” needs a reference. This comment applies here and also elsewhere in the text.

Response: We thank the reviewer for pointing this out. We have tried to make sure all the presented information can be confirmed in referenced literature. We have added extra references to the following locations: lines 34, 35, 81, 88, 100, 137, 361.

Reviewer 2 Report

The manuscript by van Dalen et al. reviews the literature about how to repolarize macrophages into the M1 subset, a population of macrophages that can potentially stimulate anti-tumor immune responses at the tumor microenvironment. This is a well-written, comprehensive manuscript with a lot of useful information for both researchers and oncologists. I only have two comments.

1. It would be helpful if the authors could state clearly whether the described experimental results (or conclusions) were derived from mouse or human studies, so that the readers can be aware of the real-life implications of these results.

2. There are several statements without supporting references. For instance, in Line 33, Page 1, the statement “TAMs are responsible for…from attacking the tumour.” needs a reference. In Line 34, Page 1, the statement “Instead, TAMs enhance tumour development…with a poor prognosis.” needs a reference. In Line 98, Page 3, “Tumour expansion is followed by…benign-to-malignant transition.” needs a reference. This comment applies here and also elsewhere in the text.

Author Response

Dear Nancy Li,

Please find enclosed our revised manuscript entitled "Molecular Repolarisation of Tumour-Associated Macrophages" (Manuscript ID: molecules-402551). We were happy to see that both reviewers were very positive about our work and we appreciate the suggestions made to improve our manuscript. We have revised the manuscript in response to the suggestions and comments by the reviewers and we have highlighted all revisions using the "Track Changes" function in Microsoft Word. Please find below our response to the reviewers.

Response to reviewer #1:

1.      However, the information on the origin of TAM is lacking. The authors should describe the origin of TAMs and the molecular mechanisms underlying their polarization into M2 state.

Response: We thank the reviewer for pointing this out. We have further clarified the origin of TAMs in the introductory section (lines 80-89) by slightly rephrasing the original text and the addition of a sentence to clarify where TAMs originate from and the molecular mechanism that determines their polarisation state. This section now reads:

“Bone marrow-derived monocytes (F4/80low) are recruited to the tumour via secretion of CSF1 and monocyte chemoattractant protein 1 (MCP1 or CCL2). These monocytes differentiate into M2-polarised macrophages under influence of CSF1, IL4 and IL13 [1,3,19].”

Several reviews have covered this subject and are referenced in the text.

2.      Moreover, some clinical studies were described intermingling with studies on mouse models. The authors should describe explicitly the types of studies, clinical or pre-clinical ones, of each citation.

Response: We thank both reviewers for this suggestion. We have reviewed and updated the text to state more clearly the type of study. Adaptations are indicated by the “track changes” function in Microsoft Word.

3.      Furthermore, the manuscript lacks the description on the molecular and pharmacological mechanisms of the agents, particularly small molecules. The authors should try to discuss the pharmacological mechanisms of small molecules as much as possible.

Response: We fully agree that the pharmacological mechanisms are a focal point of the manuscript and we endeavoured to review all the available information on the molecular targets and polarisation mechanisms of the presented TAM repolarisation molecules. Sadly, for some of the presented molecules, for instance baicalin, chlorogenic acid, emodin, hydrazinecurcumin etc., the molecular target has not yet been elucidated. In these cases, the mechanism of action that has been hypothesised by the original authors has been reviewed in the text. An overview of the polarisation pathways are presented in table 1. When the target and the signalling pathway are not known this has been stated in Table 1. We have reviewed and described all available information to the best of our knowledge.

Response to reviewer #2:

1.      It would be helpful if the authors could state clearly whether the described experimental results (or conclusions) were derived from mouse or human studies, so that the readers can be aware of the real-life implications of these results.

Response: As stated above in response to reviewer 1, we thank both reviewers for their suggestion and we have reviewed and updated the text to state more clearly the type of study. Adaptations are indicated by the “track changes” function in Microsoft Word.

2.      There are several statements without supporting references. For instance, in Line 33, Page 1, the statement “TAMs are responsible for…from attacking the tumour.” needs a reference. In Line 34, Page 1, the statement “Instead, TAMs enhance tumour development…with a poor prognosis.” needs a reference. In Line 98, Page 3, “Tumour expansion is followed by…benign-to-malignant transition.” needs a reference. This comment applies here and also elsewhere in the text.

Response: We thank the reviewer for pointing this out. We have tried to make sure all the presented information can be confirmed in referenced literature. We have added extra references to the following locations: lines 34, 35, 81, 88, 100, 137, 361.

Molecules EISSN 1420-3049 Published by MDPI AG, Basel, Switzerland RSS E-Mail Table of Contents Alert
Back to Top